# Signs and Symptoms of Acute Bowel Inflammation and the Risk of Progression to Inflammatory Bowel Disease: A Retrospective Analysis

**DOI:** 10.3390/jcm11154595

**Published:** 2022-08-06

**Authors:** Asaf Levartovsky, Tal Ovdat, Yiftach Barash, Zohar Ben-Shatach, Yael Skinezes, Stuart Jesin, Robert Klempfner, Ehud Grossman, Uri Kopylov, Shomron Ben-Horin, Bella Ungar

**Affiliations:** 1Sackler Medical School, Tel Aviv University, Tel Aviv 69978, Israel; 2Department of Gastroenterology, Sheba Medical Center, Tel Hashomer, Ramat Gan 52621, Israel; 3Leviev Heart Center, Sheba Medical Center, Tel Hashomer, Ramat Gan 52621, Israel; 4Department of Diagnostic Imaging, Sheba Medical Center, Tel Hashomer, Ramat Gan 52621, Israel; 5Deep Vision Lab, Sheba Medical Center, Tel Hashomer, Ramat Gan 52621, Israel; 6Internal Medicine Wing, Sheba Medical Center, Tel Hashomer, Ramat Gan 52621, Israel

**Keywords:** IBD progression, colitis, enteritis, bowel inflammation, prediction, imaging

## Abstract

Episodes of acute ileitis or colitis have been associated with future development of inflammatory bowel diseases (IBD). Nevertheless, the rate of future IBD among patients diagnosed with signs or symptoms of acute bowel inflammation is unknown. We aimed to assess the risk of IBD development among patients presenting with signs or symptoms of ileitis or colitis. We searched for all patients that visited the emergency department (ED) and underwent abdominal computed tomography (CT) who were eventually diagnosed with IBD during gastroenterology follow-ups within 9 years from the index admission. Multivariable models identified possible predictors of patients to develop IBD. Overall, 488 patients visited the ED and underwent abdominal imaging with abnormal findings, and 23 patients (4.7%) were eventually diagnosed with IBD (19 Crohn’s, 4 ulcerative colitis). Patients with a future IBD diagnosis were significantly younger (28 vs. 56 years, *p* < 0.001) with higher rates of diarrhea as a presenting symptom (17.4% vs. 4.1%, *p* = 0.015) compared to non-IBD patients. On multivariable analysis, age (*p* < 0.001), colitis (*p* = 0.004) or enteritis (*p* < 0.001) on imaging and a diagnosis of diarrhea in the ED (*p* = 0.02) were associated with development of IBD. Although alarming to patients and families, ED admission with intestinal inflammatory symptoms leads to eventual diagnosis of IBD in <5% of patients during long-term follow-up.

## 1. Introduction

Inflammatory bowel diseases (IBD), including Crohn’s disease (CD) and ulcerative colitis (UC), are disorders characterized by chronic inflammation of the gastrointestinal tract. Presenting symptoms often include abdominal pain, diarrhea or bloody stools [1,2]. Most cases of IBD are diagnosed after patients seek medical attention following abdominal symptoms. Nevertheless, the final diagnosis can be made even months or years after symptoms emerge [3,4,5]. Acute ileitis or colitis may represent the first manifestation of IBD, although these presentations can be secondary to other etiologies such as infections, drugs, ischemia or rheumatologic conditions [6,7,8]. Furthermore, episodes of acute intestinal infections have been associated with future development of IBD [9,10,11]. A few studies suggest that in patients with acute terminal ileitis evident on colonoscopy, the presence of symptoms such as abdominal pain were associated with a higher likelihood of eventual development of CD compared to those without gastrointestinal symptoms [12,13]. Median follow-up time between presentation of acute ileitis to development of CD has been reported between a range of 7.5 and 43.2 months from index endoscopy [10]. While colonoscopy with intestinal biopsies demonstrating features of chronicity is the basis for diagnosis, imaging is also a fundamental pillar of IBD diagnosis [14,15,16]. Markers such as C-reactive protein (CRP) can be sensitive markers of systemic inflammation [17,18]. In patients with chronic abdominal symptoms, this marker can assist in differentiating IBD from normal individuals or from patients with functional bowel disorders [19]. An additional marker, fecal calprotectin, can have diagnostic yield in IBD, yet it is less useful in the setting of an ED admission [17,20]. The rate of subsequent development of IBD in patients who presented to ED with acute bowel symptomology and abnormal CT findings remains little explored. This knowledge gap often creates unnecessary anxiety among the patients and their families and also creates uncertainty among the caring physician as to whether any closer long-term follow up for IBD is needed in this specific patient group. Therefore, the current study aimed to assess the risk for a future development of IBD in a large cohort of patients presenting to a large tertiary hospital ED with signs of acute bowel inflammation on abdominal imaging.

## 2. Materials and Methods

### 2.1. Patient Population and Design

This was a retrospective cohort study of all patients aged 70 or younger who were admitted (index visit) to the emergency department (ED) at Sheba Medical Center, a tertiary academic center in Israel, with symptoms that could be attributed to IBD (abdominal pain, diarrhea, fever) and who, as part of their work-up, had undergone an abdominal CT at the ED with abnormal findings reported, between 1 January 2011 and 31 December 2020.

Abnormal CT findings were defined as containing any of the following: wall thickening, fat infiltration, free fluid, enlarged lymph nodes, colitis, obstruction, enteritis, diverticulitis, appendicitis, vascular congestion, stricture or space- occupying-lesion.

### 2.2. Exclusion Criteria

Patients over 70 years old, as new onset IBD at that age group is rare;Patients not presenting with abdominal symptoms described above;Previous diagnosis of IBD or cancer;Abdominal CT scan with no abnormal findings;Patients with a presumptive diagnosis of IBD in the index hospitalization but without any further confirmation of an IBD diagnosis during follow up;Patients without any follow-up visit in the gastroenterology department in Sheba medical center after the index hospitalization;Patients who died during the index hospitalization.

### 2.3. Data Collection

For each patient included, the following data were retrieved from the computerized database: reason for referral, age and demographic characteristics, background diseases, laboratory parameters, hospitalization outcomes and discharge diagnoses. For patients with a future IBD diagnosis, the electronic medical records were manually reviewed for clinical data, including IBD type, disease extent, behavior, extraintestinal manifestations and therapies. Specific radiological parameters were defined and extracted from CT scans based on imaging radiologists’ reports, each of them revised by senior abdominal radiologists. To avoid redundancy from repeated visits of the same patient, all visits but the first ED visit were excluded.

The primary analysis examined patients who met the inclusion criteria and had a follow-up visit in the gastroenterology department at Sheba medical center. An additional analysis, including a cohort of 1551 patients, did not exclude patients without a gastroenterology follow-up.

### 2.4. Future Development of IBD

We searched the computerized database for patients diagnosed with CD or UC during a visit to the gastroenterology department within a maximal follow-up time of 9 years from the index hospitalization to the ED. The outcome analyzed was an established diagnosis of IBD determined by a senior gastroenterologist based on endoscopic, histological and imaging parameters.

### 2.5. Statistycal Analysis

Descriptive statistics were employed. Continuous variables were presented as medians and intra-quartile ranges (IQR) and categorical variables as medians and percentages. To compare between the sub-groups developing subsequent IBD or not, chi-square for categorical variables and *t*-tests or Mann–Whitney–Wilcoxon tests were employed as appropriate for normally/non-normally distributed continuous variables. Normality of continuous variables was assessed using the Shapiro–Wilk test. In order to explore the relationship between patients’ characteristics and the outcome of developing IBD, a Cox proportional hazards model was performed, which takes into account the competing risk of a mortality event and produces HR (95% CI) accordingly.

Pre-specified covariates were age, gender, WBC and hemoglobin (in the ED or first measurement in admission), diarrhea and abdominal pain as a presenting symptom in the ED. In addition, variables that were significantly associated at the univariable analysis and without missing data were included.

All statistical tests were two-sided, and a *p* value < 0.05 was considered statistically significant. Statistical analysis was performed using R, version 4.0.3 (R Foundation for Statistical Computing, Vienna, Austria. URL https://www.R-project.org/, accessed on 3 March 2020).

### 2.6. Study Ethics and Patient Consent

This study was carried out in accordance with the ethical guidelines of the Declaration of Helsinki. The study was approved by the institutional review board of the Sheba Medical Center. Since this was a retrospective analysis, no informed consent was obtained.

## 3. Results

### 3.1. Study Cohort

Overall, 25,282 patients visited the ED and underwent abdominal CT with intravenous contrast injection throughout the study period. We included only the first ED visit for each patient. We excluded 8195 patients with a prior history of IBD or malignancy and 4937 patients above the age of 70. An additional 9537 patients were excluded due to complaints and symptoms not relevant to the gastrointestinal tract. Patients with no pathological findings on abdominal CT (1575) were excluded as well. Of the 1038 remaining patients, we excluded patients who were diagnosed with IBD during the index hospitalization yet had no confirmation of diagnosis during follow-up in the gastroenterology department (to exclude cases of erroneous diagnosis/previous IBD) (n = 106). In addition, patients who died during the index hospitalization (n = 7) and patients without any follow-up data were excluded (n = 437). The remaining study population included 488 patients who visited the ED with relevant complaints and underwent an abdominal CT with abnormal findings (Figure 1). The median age of the study population was 55 years (IQR 40.75–63 years). Of these, 255 patients (52.3%) were males, and 104 patients (21.3%) were discharged from the ED without hospitalization. The radiographic parameters that were most commonly reported on abdominal CT scans were bowel wall thickening (47.5%), fat infiltration (29.7%) and free fluid (28.1%). These radiographic parameters are outlined in Figure 2.

### 3.2. From Acute Intestinal Inflammation Progression to IBD

Of the 488 patients comprising the study population, 23 patients (4.7%) were eventually diagnosed with IBD. All patients had a follow-up visit at the gastroenterology department at Sheba medical center after the index visit. Patients who were eventually diagnosed with IBD had an initial follow-up visit within a median of 7 days (IQR 4–15) compared to 46 days (IQR 7–305) in the group of patients who did not develop IBD (*p* < 0.001). In both groups, a similar proportion of patients (383/465, 82.4% of non-IBD patients and 19/23, 82.6% of IBD patients) had an additional visit at the gastroenterology department. There was no difference in median follow-up time from the index visit in the IBD group (6.23 years, IQR 4.17–6.92) compared to the non-IBD group (5.92 years, IQR 3.28–7, *p* = 0.824).

CD was diagnosed in 19 patients, and 4 patients were diagnosed with UC. Patients with a future IBD diagnosis were significantly younger (median 28 years, IQR 20.5–43, versus 56 years, IQR 43–63, *p* < 0.001) and had a lower BMI (body mass index, median 21.23 kg/m, IQR 19.65–22.32, versus 26.12 kg/m, IQR 22.76–29.97, *p* < 0.001) compared to non-IBD patients at ED admission. Diarrhea was the presenting symptom significantly more often among the former group at ED admission (17.4% vs. 4.1%, *p* = 0.015) and upon discharge from hospitalization (27.3% vs. 9.4%, *p* < 0.001). There were no differences in laboratory data (white blood cells, hemoglobin, CRP, albumin) between the groups on ED presentation. Twenty-three patients progressed to IBD with a median of 60 days (IQR 30–120 days) from the time of the ED admission (Appendix A). The clinical characteristics of patients who progressed to IBD and of non-IBD patients are detailed in Table 1.

Of the 23 patients with subsequent IBD, 19 developed CD, of whom the majority (63.1%) had a non-penetrating and non-stricturing disease with ileal involvement. Six patients (26%) had extra-intestinal manifestations, and three patients (13%) were current or past smokers. Nine patients (39.1%) were treated with steroids (prednisone and/or budesonide) and five patients (21.7%) received biologic therapy (infliximab, adalimumab or vedolizumab) upon diagnosis of IBD (Table 2).

Several radiographic parameters were proportionally higher on abdominal imaging of patients with a future diagnosis of IBD (Table 1). These included ascending colitis (6/23, 26.1% versus 22/488, 4.7%, *p* < 0.001), transverse colitis (6/23, 26.1% versus 15/488, 3.2%, *p* < 0.001), descending colitis (7/23, 30.4%, versus 32/488, 6.9%, *p* < 0.001), pan-colitis (4/23, 17.4% versus 11/488, 2.4%, *p* = 0.001) and enteritis (5/23, 21.7% versus 31/488, 6.7%, *p* = 0.022). IBD was diagnosed in 28.6% (6/21) of patients with transverse colitis on imaging and in 26.6% (4/15), 21.4% (6/28), 17.9% (7/39) and 13.9% (5/36) of patients with pan-colitis, ascending colitis, descending colitis and enteritis, respectively. Figure 3 presents the frequency of patients that developed IBD among subsets of patients, each subset having a different radiographic parameter.

On a univariable analysis, age (per 5 years, HR 0.92, CI 0.89–0.95, *p* < 0.001], colitis on imaging (HR 5.43, CI 2.35–12.5, *p* < 0.001), enteritis on imaging (HR 3.59, CI 1.33–9.67, *p* = 0.011) and a diagnosis of diarrhea at the ED (HR 4.58, CI 1.56–13.4, *p* = 0.006) were associated with developing IBD. Specific locations of colitis on imaging—descending (HR 5.3, CI 2.17–12.89, *p* < 0.001), transverse (HR 8.8, CI 3.47–22.4, *p* < 0.001) and ascending colitis (HR 6.09, CI 2.4–15.46, *p* < 0.001)—were associated as well with development of IBD. There was a trend toward higher rates of fat infiltration on imaging of the non-IBD group (HR 0.34, CL 0.1–1.15, *p* = 0.08). On a multivariable cox regression model with death as a competing risk, age (per 5 years, HR 0.64, CI 0.53–0.77, *p* < 0.001), colitis on imaging (HR 3.35, CI 1.48–7.6, *p* = 0.004), enteritis on imaging (HR 7.3, CI 2.48–21.44, *p* < 0.001) and a diagnosis of diarrhea in the ED (HR 4.79, CI 1.42–16.18, *p* = 0.02) were associated as well with the development of IBD (Figure 4).

A second analysis was performed for all patients who visited the ED due to abdominal symptoms, not limited to those with a follow-up visit in the medical center. Overall, 25,282 patients visited the ED and underwent abdominal CT. After excluding those with a prior diagnosis of cancer, IBD or age above 70, a total of 1551 patients were included with a future IBD rate of 1.5% (23/1551). In this analysis, the abdominal imaging parameters associated with a future diagnosis of IBD were similar to the original cohort (ascending colitis, transverse colitis, descending colitis, pan-colitis and enteritis). In addition, age (*p* < 0.001) and diarrhea as a presenting symptom (*p* < 0.001) were significantly associated with a future diagnosis of IBD. On multivariate analysis, age (*p* < 0.001) and colitis on imaging (*p* = 0.001) were associated with the development of IBD.

## 4. Discussion

Diagnosing IBD based on a single episode of acute ileitis or colitis is challenging due to the chronic histopathological pattern of the disease necessary for the diagnosis. Therefore, defining risk factors for the likelihood of a future diagnosis of IBD is of clinical importance. This can assist us in defining the optimal follow-up of patients following an acute episode of bowel inflammation. In our study, approximately 4.7% of patients who presented to the ED with related symptoms and underwent abdominal CT with abnormal findings developed IBD in up to a nine-year follow-up time from the index visit. The progression rates reported by previous studies were higher; nevertheless, these were assessed in asymptomatic patients with incidentally diagnosed terminal ileitis on endoscopies. Zhang and colleagues reported a 20% (7/34) progression rate to overt CD [12]. Two additional studies reported 9% (3/32) and 14% (6/43) progression rates to CD, with a median follow up time of 38.4 months (range 1–61) and 40 months (range 4–106), respectively [9,21]. However, all the above studies explored progression of asymptomatic patients with isolated terminal ileitis on endoscopies, whereas our study examined patients with an acute episode of characteristic clinical and radiographic manifestations. Tse and colleagues examined patients undergoing diagnostic ileocolonoscopy due to unexplained gastrointestinal symptoms or abnormal abdominal imaging findings. Among these patients with endoscopic isolated acute terminal ileitis, 4.6% of patients were eventually diagnosed with CD [10]. Although the selected cohort and the progression rates were relatively similar to our study, the additional diagnostic tier of endoscopy and histology cannot be overlooked. A multi-center retrospective study by Cantoro and colleagues explored the time delay from symptom onset to definitive diagnosis [3]. The median period of diagnostic delay was three months, with an interquartile range similar to our study, 30–120 days. Thus, the data presented in this work and in our study suggest that individuals presenting to the ER with imaging-detected ileitis or colitis have very low risk of developing IBD in ensuing years if their diagnosis is not established within several months of the index visit. Hence, in our view, specific routine work-up, aside from repeat laboratory and clinical work-up, is probably not necessary. In addition, the relatively short period of time between the acute bowel inflammation and IBD diagnosis in our cohort could probably also be attributed to availability of gastroenterological follow-up and awareness of the ED personnel in a tertiary medical center. These factors could vary in different medical centers.

In our study, patients with a future diagnosis of IBD were younger than control patients. Tse and colleagues demonstrated that among patients with isolated acute terminal ileitis, the patients that progressed to CD were older than those without CD (median 56.8 vs. 43.9 years, respectively) [10]. In contrast, in another study describing 105 patients with a first episode of colitis, the mean age of the patients with infectious colitis was 45 years, compared to the IBD group that was younger (38 years) [22]. This finding is consistent with our findings. No association was demonstrated between gender or comorbidities and progression to IBD. There were no significant differences in laboratory data between the groups on ED presentation. Nonetheless, a trend for higher CRP values was demonstrated among patients who progressed to IBD. This goes in line with the fact that elevated inflammatory markers are associated with IBD flares [18,19]. The fact that other acute abdominal conditions (such as diverticulitis and appendicitis) are also associated with these markers probably contributed to the borderline statistical significance.

As the main objective of this study was to identify predictors of a future IBD diagnosis, imaging at disease presentation was an integral component of this analysis. Tse and colleagues reported the presence of stricturing or narrowing on imaging as a predictor of eventual CD development. In our study, these radiographic features were somewhat sparsely present in imaging reports of patients who developed IBD in the future (4.3% obstruction, 8.7% partial obstruction, 4.3% strictures). Nevertheless, several radiographic parameters including colitis (ascending, transverse, descending, pan-colitis) and enteritis were indeed significantly more common in future IBD cases when using the cox regression models.

In the second analysis, we identified predictors of a future IBD diagnosis in a cohort of 1551 patients not limited to those with a follow-up visit in the medical center. As expected, this analysis demonstrated a lower progression rate to IBD (1.5%) among patients included in the study. However, the imaging and clinical features associated with progression to IBD were similar to the first analysis, emphasizing the role of these parameters in predicting a future IBD diagnosis.

We are aware of several limitations in our study. First, the data collected in this study refer to patients who visited our tertiary medical center and visited the gastroenterology department afterwards at least once. There is a possibility that patients in the non-IBD group might have been eventually diagnosed with IBD in another medical center. To overcome this limitation, we limited our cohort to only those patients returning to our gastroenterology institute for any reason during the follow-up period. In addition, we demonstrated that the overwhelming majority of patients had a further visit to the medical center and that in both groups the overall follow-up time was similar. Therefore, the possibility of an IBD diagnosis in another medical center is less likely. Notwithstanding, as suggested by the second analysis, the IBD rate could in fact be lower as many of the patients not returning to our medical center have not developed IBD. Secondly, we excluded patients without abdominal imaging. As a result, only patients with presenting symptoms or laboratory data grave enough to clinically justify imaging were included. However, imaging has a prominent role in the toolbox needed for diagnosing IBD; therefore, we opted to integrate imaging features in our analysis. Ordering CT at the ED probably marks a more severely symptomatic group of patients, in whom the patient’s and caregiver’s concern for need of future follow-up or possibility of IBD is likely more clinically relevant.

## 5. Conclusions

In conclusion, in this study we found that even among patients presenting to the ED with inflammatory intestinal symptoms and with abnormal findings on abdominal CT, the overall risk for later development of IBD is very low. This should reassure physicians and patients about the long-term outcome of such disconcerting episodes. Furthermore, we identified clinical and radiographic features associated with development of IBD, which can assist decision making for clinicians in the ED and from which patients could benefit from a designated monitoring protocol following an acute episode.

## Figures and Tables

**Figure 1 jcm-11-04595-f001:**
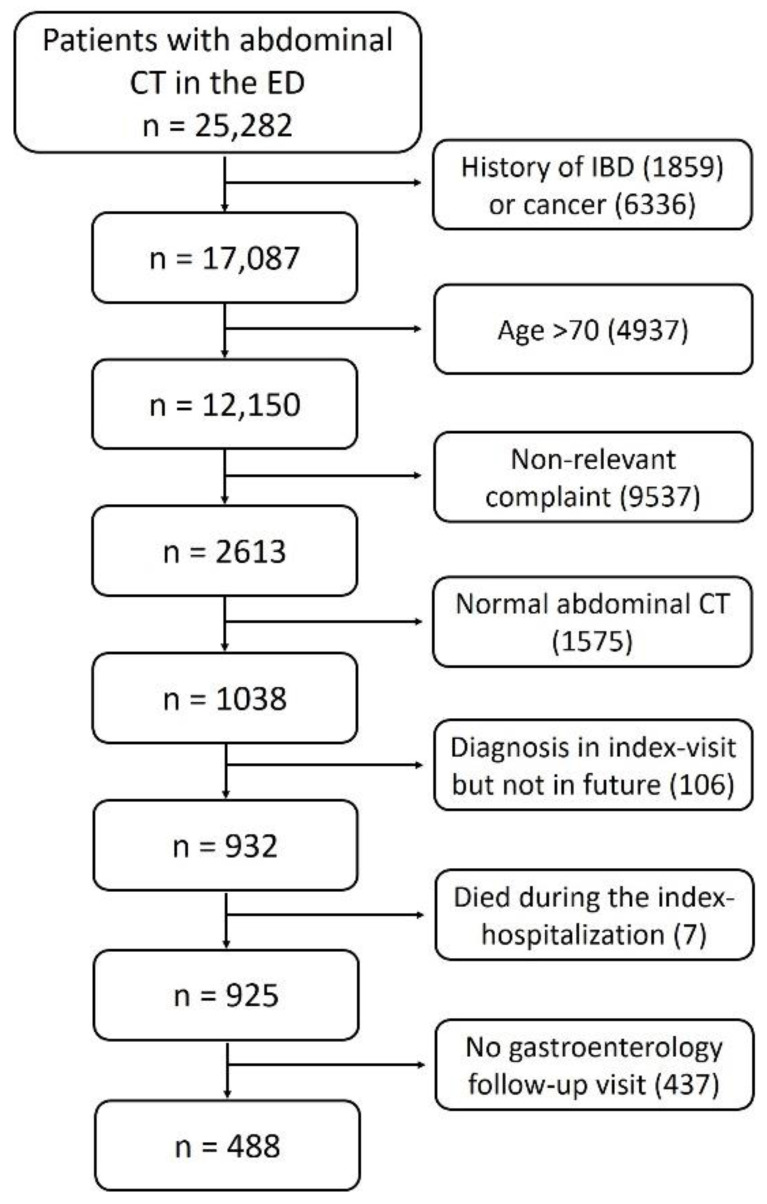
Flow chart showing the screened, included and excluded patients (ED, emergency department; IBD, inflammatory bowel disease).

**Figure 2 jcm-11-04595-f002:**
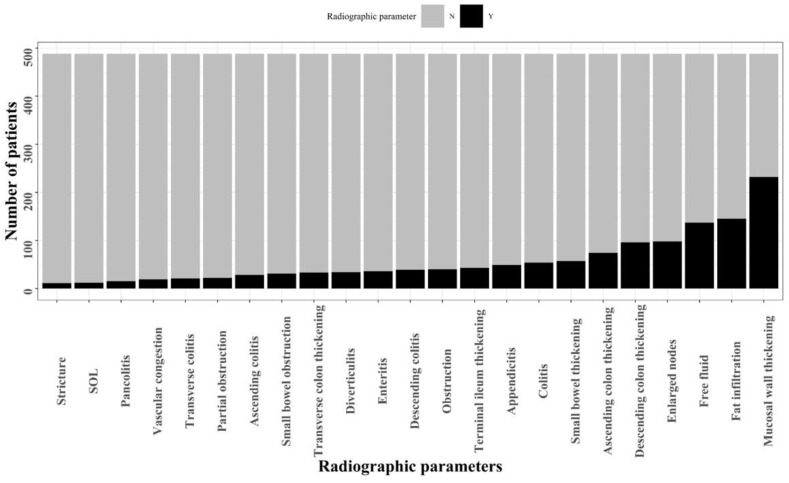
Frequency of radiographic parameters as assessed by abdominal CT (SOL—space-occupying lesion).

**Figure 3 jcm-11-04595-f003:**
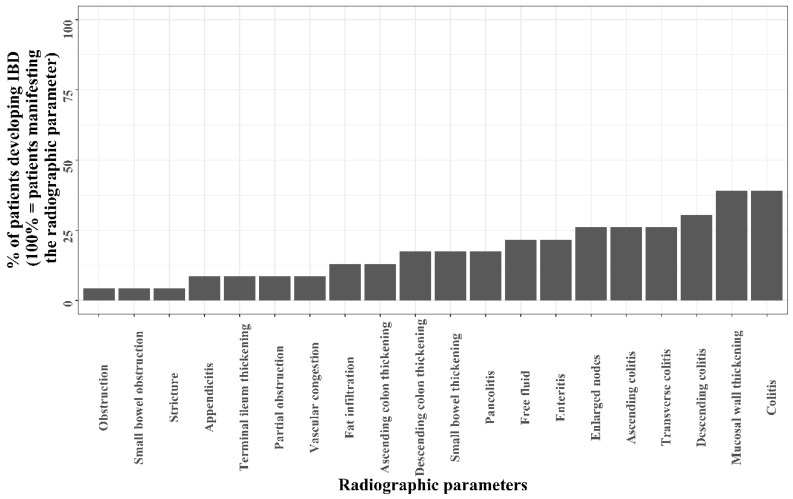
Frequency of the patients developing IBD as per different radiographic parameters.

**Figure 4 jcm-11-04595-f004:**
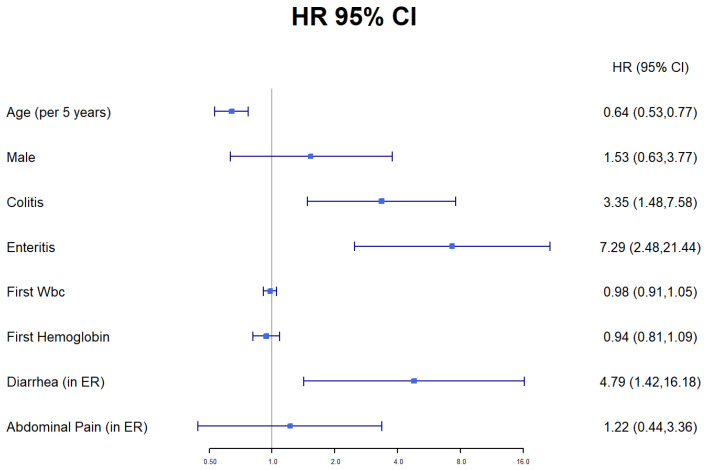
Multivariable analysis of factors associated with developing IBD. The only significant factors were age, diarrhea on presentation and enteritis or colitis on imaging (WBC—white blood cells).

**Table 1 jcm-11-04595-t001:** Characteristics of study population—demographics, comorbidities, clinical and imaging features at ED admission.

	Non-IBD (n = 465)	Future IBD (n = 23)	*p*-Value
Male gender, n (%)	243 (52.3)	12 (52.2)	1
Age (median, IQR)	56 (43, 63)	28 (20.5, 43)	**<0.001**
BMI (median, IQR)	26.12 (22.76, 29.97)	21.23 (19.65, 22.32(	**<0.001**
Comorbidities, n (%)			
Hypertension, n (%)	67 (14.4)	0 (0)	0.1
Diabetes mellitus, n (%)	43 (9.2)	0 (0)	0.25
Anemia, n (%)	48 (10.3)	1 (4.3)	0.56
Past smoker, n (%)	29 (6.2)	0 (0)	0.43
Laboratory features (IQR)			
WBC	10.88 (7.96, 14.23)	9.78 (8.21, 13.44)	0.595
CRP	34.38 (7.45, 104.28(	67.42 (49.16, 145.8)	0.07
Albumin	3.6 (3.3, 4 (	3.4 (3.1,3.7)	0.210
Hemoglobin	13.01 (11.49, 14.28(	12.49 (11.27, 13.32)	0.364
ED admission, n (%)			
Fever (median, IQR)	36.8 (36.6, 37.2(	36.9 (36.75, 37.3)	0.214
Diarrhea diagnosis	19 (4.1)	4 (17.4)	**0.015**
Abdominal pain diagnosis	104 (22.4)	9 (30.4)	0.52
ED discharge	103 (22.2)	1 (4.3)	0.076
Hospitalization, n = 384 (%)			
Re-hospitalizations—30 days	79 (17)	7 (30.4)	0.172
Re-hospitalizations—90 days	118 (25.4)	8 (34.8)	0.45
Surgery during index hospitalization	83 (22.9)	2 (9.1)	0.21
Imaging features, n (%)			
Ascending colitis	22 (4.7)	6 (26.1)	**<0.001**
Transverse colitis	15 (3.2)	6 (26.1)	**<0.001**
Descending colitis	32 (6.9)	7 (30.4)	**<0.001**
Pan-colitis	11 (2.4)	4 (17.4)	**0.001**
Enteritis	31 (6.7)	5 (21.7)	**0.022**

BMI—body mass index; CRP—C-reactive protein; ED—emergency department; IQR—interquartile range; WBC—white blood cells. *p* value < 0.05 was considered statistically significant (highlighted in bold text).

**Table 2 jcm-11-04595-t002:** IBD characteristics of patients.

Number of Patients	23
Male, n (%)	11 (47.8)
Crohn’s disease, n (%) Ulcerative colitis, n (%)	19 (82.6) 4 (17.4)
Current or past smoker, n (%	3 (13)
EIM, n (%)	6 (26)
CD extent at diagnosis, n (% of CD patients) L1 (ileal) L2 (colonic) L3 (ileo-colonic)	12 (63.1) 1 (5.3) 6 (31.6)
CD behavior at diagnosis, n (% of CD patients) B1 (non-stricturing and non-penetrating) B2 (stricturing) B3 (penetrating)	12 (63.1) 4 (21.1) 3 (15.8)
UC extent, n (%) E1 (proctitis) E2 (left-sided colitis) E3 (right-sided colitis)	1 (25) 2 (50) 1 (25)
Therapy on diagnosis, n (%) Steroids Immunomodulators Biologics ASA	9 (39.1) 2 (8.7) 5 (21.7) 3 (13)

ASA—5-aminosalicylic acid; EIM—Extra-intestinal manifestation; IQR—Interquartile range.

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
