# Peer review of "Signs and Symptoms of Acute Bowel Inflammation and the Risk of Progression to Inflammatory Bowel Disease: A Retrospective Analysis"

_jcm, 2022, doi:10.3390/jcm11154595_

Round 1

Reviewer 1 Report

In this article, authors nicely described the risk factors and possible diagnostic procedures for intestinal inflammation and progression of inflammatory bowel disease. They assessed the risk of future IBD development among patients with ileitis and colitis syndrome. Overall, the article demonstrates significant findings and useful analysis for the readers. Please do some minor changes as suggested bellow:

1. Enlarge and bold the words font size  in figure 2 and 3, hard to read.

2. Consider adding more citations to validate the descriptions, citation number is low.

Reviewer 2 Report

The manuscript is regarding to “acute intestinal inflammation” and then diagnosed as IBD. The idea is interesting, and could provide clinical benefit. However, there are points to be clarified.

1.      the hypothesis of the study would be clearly revised. IBD is a chronic disease, and most diagnosed since young age. The authors mentioned that follow up within 9 yrs. IBD diagnosis was “missed” more than “developed” from ileitis or enteritis. Thus, the authors should change the title and clarify the hypothesis through the introduction as well as abstract, keywords and discussion. The authors may consider “acute bowel inflammation symptoms and signs……” instead of “acute intestinal inflammation” or “IBD with initial presentation of acute bowel inflammatiory…”

2.      How many % of patients received endoscopy, including colonoscopy in the initial acute inflammatory ER visit. Why the patients received endoscopy check or not if CT showing enteritis.

3.      the follow up period is mentioned in abstract, but not in method, the authors may mention it in the method section of the text. However, a Kaplan-meier curse may be considered.

4.      In table 2, hospitalization and surgery were analysis. Did patients’ hospitalization and surgery related to abdomen. If not, both factors were not valuable for this study and both variables may be deleted.

5.      Image finding were important variable as the authors concluded. However, these variables were not presented in table 2. Besides, the criteria image finding about ileitis or enteritis on CT would be mentioned.

6.      In table 1, 1575 patients with “normal “ CT were excluded in this study.  A “normal” CT did not assure a IBD diagnosed in the follow up, a a selection bias may be mentioned. Did the authros have data about subsequently diagnosed as IBD in these “normal” CT patients.

7.      Multiple variable analysis could be related to univariable analysis. Thus, the figure 4 should be base on table 2. Table 1 is not doubled and should be combined to table 2.

8.      The study ethic should be mentioned, and possible certification of Institutional review board would be provided.
